# LXRα Regulates ChREBPα Transactivity in a Target Gene-Specific Manner through an Agonist-Modulated LBD-LID Interaction

**DOI:** 10.3390/cells9051214

**Published:** 2020-05-13

**Authors:** Qiong Fan, Rikke Christine Nørgaard, Ivar Grytten, Cecilie Maria Ness, Christin Lucas, Kristin Vekterud, Helen Soedling, Jason Matthews, Roza Berhanu Lemma, Odd Stokke Gabrielsen, Christian Bindesbøll, Stine Marie Ulven, Hilde Irene Nebb, Line Mariann Grønning-Wang, Thomas Sæther

**Affiliations:** 1Department of Molecular Medicine, Institute of Basic Medical Sciences, Faculty of Medicine, University of Oslo, N-0317 Oslo, Norway; qiong.fan@medisin.uio.no (Q.F.); kristin.vekterud@medisin.uio.no (K.V.); christian.bindesboll@gmail.com (C.B.); 2Department of Nutrition, Institute of Basic Medical Sciences, Faculty of Medicine, University of Oslo, N-0317 Oslo, Norway; rikkechristine@hotmail.com (R.C.N.); ceciliem.ness@gmail.com (C.M.N.); christin.zwafink@gmail.com (C.L.); helensoedling@gmail.com (H.S.); jason.matthews@medisin.uio.no (J.M.); smulven@medisin.uio.no (S.M.U.); h.i.nebb@medisin.uio.no (H.I.N.); lmgronningwang@gmail.com (L.M.G.-W.); 3Department of Informatics, Faculty of Mathematics and Natural Sciences, University of Oslo, N-0317 Oslo, Norway; ivargry@student.matnat.uio.no; 4Department of Biosciences, Faculty of Mathematics and Natural Sciences, University of Oslo, N-0317 Oslo, Norway; r.b.lemma@ncmm.uio.no (R.B.L.); o.s.gabrielsen@ibv.uio.no (O.S.G.)

**Keywords:** glucose, cholesterol, lipid metabolism, nuclear receptors, liver, trans-coactivation, ChIP

## Abstract

The cholesterol-sensing nuclear receptor liver X receptor (LXR) and the glucose-sensing transcription factor carbohydrate responsive element-binding protein (ChREBP) are central players in regulating glucose and lipid metabolism in the liver. More knowledge of their mechanistic interplay is needed to understand their role in pathological conditions like fatty liver disease and insulin resistance. In the current study, LXR and ChREBP co-occupancy was examined by analyzing ChIP-seq datasets from mice livers. LXR and ChREBP interaction was determined by Co-immunoprecipitation (CoIP) and their transactivity was assessed by real-time quantitative polymerase chain reaction (qPCR) of target genes and gene reporter assays. Chromatin binding capacity was determined by ChIP-qPCR assays. Our data show that LXRα and ChREBPα interact physically and show a high co-occupancy at regulatory regions in the mouse genome. LXRα co-activates ChREBPα and regulates ChREBP-specific target genes in vitro and in vivo. This co-activation is dependent on functional recognition elements for ChREBP but not for LXR, indicating that ChREBPα recruits LXRα to chromatin in *trans*. The two factors interact via their key activation domains; the low glucose inhibitory domain (LID) of ChREBPα and the ligand-binding domain (LBD) of LXRα. While unliganded LXRα co-activates ChREBPα, ligand-bound LXRα surprisingly represses ChREBPα activity on ChREBP-specific target genes. Mechanistically, this is due to a destabilized LXRα:ChREBPα interaction, leading to reduced ChREBP-binding to chromatin and restricted activation of glycolytic and lipogenic target genes. This ligand-driven molecular switch highlights an unappreciated role of LXRα in responding to nutritional cues that was overlooked due to LXR lipogenesis-promoting function.

## 1. Introduction

Glucose and lipid metabolism are tightly connected and coordinately regulated in mammals to maintain whole-body energy homeostasis. In the liver, excess dietary carbohydrates are converted to fatty acids through de novo lipogenesis (DNL) destined for long-term storage as triglycerides in adipose tissue. Dysregulation of lipogenesis contributes to non-alcoholic fatty liver disease (NAFLD), which is associated with increased risk of metabolic syndrome and type 2 diabetes [1,2]. Several transcription factors (TFs) play essential roles in modulating glucose and lipid metabolism, including the cholesterol-sensing nuclear receptor liver X receptor (LXR) and the glucose-sensing TF carbohydrate responsive element-binding protein (ChREBP) [3,4].

While showing high sequence homology, the two LXR subtypes differ in their distribution (reviewed in Reference [5]): LXRα is predominantly expressed in metabolically active tissues, for example, liver and adipose tissue, while LXRβ is ubiquitously expressed [6]. Both isoforms heterodimerize with retinoid X receptors (RXRs) and regulate expression of genes involved in cholesterol homeostasis, lipid and glucose metabolism and inflammation [5,7,8,9,10], by binding to LXR response elements (LXREs; two direct repeats AGGTCA spaced by four nucleotides, DR4 elements) in gene regulatory regions [11]. in vivo LXR transactivity is modulated by oxysterols (oxidized cholesterol derivatives), which bind to the ligand-binding domain (LBD) of LXR [12]. This elicits conformational changes in LBD which leads to the release of co-repressors, recruitment of co-activators and initiation of target gene transcription [13,14,15]. By functioning as a ‘metabolic sensor,’ LXR can integrate metabolic signals into complex transcriptional responses. In the liver, LXR responds to cholesterol, insulin and glucose in the form of O-linked β-*N*-acetylglucosamine (O-GlcNAc) [12,16,17,18], by activating the transcription of ChREBP and sterol regulatory element-binding protein (SREBP)-1c [19,20], which alone or together with LXR induce the transcription of glycolytic and lipogenic enzymes, such as liver pyruvate kinase (*Lpk*), acetyl-CoA carboxylase (*Acc*), fatty acid synthase (*Fasn*) and stearoyl-CoA desaturase-1 (*Scd1*) [20,21,22]. This leads to increased *de novo* synthesis of fatty acids. The discovery of the O-GlcNAc modification of LXR [18] and its activating effects resolved a controversial issue sparked by Mitro et al. [23], proposing that glucose, as a hydrophilic molecule, could act as a direct agonistic ligand for LXR. Several synthetic ligands targeting LXR have been developed, such as the nonsteroidal agonists GW3965 and T0901317 [8,24]. However, none of these have so far reached the clinic, as one of the concerns has been LXR-induced lipogenesis [25,26].

ChREBP is a glucose-activated TF that belongs to the basic helix-loop-helix leucine zipper (bHLH/Zip) family [27]. ChREBP and its obligate partner Max-like protein X (MLX) heterodimerize via the bHLH/ZIP domain of ChREBP (illustrated in Figure 4A) [28] and bind to the conserved consensus sequence carbohydrate response element (ChoRE) located in the promoter region of glucose-responsive genes [29]. The ChoRE is comprised of two E-box (CACGTG) or E-box-like sequences spaced by five nucleotides [30]. The dominant isoform ChREBPα contains a glucose-sensing module (GSM) in its N-terminus, comprised of a low glucose inhibitory domain (LID) and a glucose response activation conserved element (GRACE) (illustrated in Figure 4A) [31]. Under low glucose condition, ChREBPα transactivity is restrained by an intramolecular inhibitory mechanism, involving LID and GRACE [32,33]. Once intracellular glucose levels increase, the inhibition is relieved, either through the direct binding of one or more glucose metabolites to LID, and/or the recruitment of co-regulatory proteins activating ChREBPα [33,34,35,36]. In contrast, the shorter isoform ChREBPβ, which is transactivated by ChREBPα through an alternative promoter 17 kb upstream of the *Chrebpα* transcription start site (TSS), lacks most of LID (the first 177 amino acids), has escaped glucose regulation and acts constitutively independent of glucose concentration [37]. In addition, glucose activates ChREBP via O-GlcNAc modification, leading to increased ChREBP transcriptional activity and recruitment to target gene promoters [38,39].

We previously reported that LXR deficiency leads to reduced ChREBP activity, resulting in reduced hepatic expression of ChREBP-specific target genes *Pklr* (*Lpk*) and *Mlxipl* (*Chrebpβ*) and less ChREBP recruitment to the *Lpk* promoter [16]. Moreover, LXRα is essential in regulating ChREBP activity in the livers of mice fed a high-glucose diet [40]. Recently, it was also shown that LXR regulates the expression of *Lpk* in the livers of mice fed an oleic acid-enriched diet [41]. It is, however, not clear whether LXR regulates ChREBP-specific target genes indirectly by activating *Chrebpα* expression or directly by modulating ChREBPα activity.

In the current study, we demonstrate that LXRα and ChREBPα interact physically and show a high co-occupancy at regulatory regions in the mouse genome. Moreover, LXRα regulates ChREBPα transactivity in a target gene-specific manner through an agonist-modulated LBD-LID interaction, where LXRα ligand binding restricts the activation of glycolytic and lipogenic target genes. We speculate that this novel function of LXRα as a ligand-driven molecular switch for ChREBP, has been overlooked due to LXR’s role in promoting DNL.

## 2. Materials and Methods

### 2.1. Materials

Formaldehyde (F1635), the synthetic LXR agonist GW3965 (G6295), dimethyl sulfoxide (DMSO; D4540), Dulbecco’s Modified Eagle’s Medium (DMEM; D6546), fetal bovine serum (FBS) (F7524), L-glutamine (G7513), penicillin-streptomycin (P4458), insulin (I9278) and D-(+)-glucose solution (G8769) were purchased from Sigma-Aldrich (St. Louis, MO, USA). Tularik (T0901317) was from Enzo Life Sciences (Farmingdale, NY, USA). DMEM, no glucose (11966-025) was purchased from Gibco, Thermo Fisher Scientific (Waltham, MA, USA). Dual Luciferase® reporter assay system (E1960) was purchased from Promega (Madison, WI, USA). All other chemicals were of the highest quality available from commercial vendors.

### 2.2. ChIP-seq Data Analysis

To generate a genome-wide map of ChREBP and LXR binding sites, two published ChIP-seq datasets were reanalyzed—ChREBP ChIP-seq data from mouse liver (C57Bl/6J male mice were fasted and high-carbohydrate refed to maximize ChREBP chromatin occupancy) [42] and LXR ChIP-seq data from mouse liver (NCBI GSE35262; C57Bl/6 female mice were treated with LXR agonist (T0901317, 30 mpk, 14 days) and anti-pan-LXR polyclonal antibody was used to capture LXRα/β) [43].

Reads were mapped to the mm9 reference genome (Assembly number: MGSCv37) using *BWA aln* (v.0.7.17) with default parameters and alignments with mapping quality less than 30 were discarded using *Samtools filter* (v.1.9). Peaks were called using *MACS* (v.2.1.1) with default parameters. Peaks within mm9 blacklisted regions [44] were discarded. There were 48 647 ChREBP peaks detected and the top 20 000 peaks with highest score were chosen for further analysis. There were 24 728 LXR peaks detected, all of which were chosen for further analysis.

Pairs of close ChREBP and LXR peaks were selected by running *bedtools closest* (v.2.26.0) on the peak summits of the ChREBP and LXR peaks and discarding pairs of close peaks with summit-to-summit distance larger than 1000 base pairs. These pairs were linked to their nearest gene by running *Rgmatch* [45] with *--report gene --distance 25 --promoter 5000* as parameters.

Genomic co-occurrence of the predicted binding sites of the different transcription factors (Figure 1G) was measured by the Forbes coefficient (https://www.ideals.illinois.edu/handle/2142/55240) using the Genomic HyperBrowser [46].

### 2.3. Animals and Fasting-Refeeding Experiments

LXRαβ wild type, LXRα^-/-^, LXRβ^-/-^ and LXRα^-/-^β^-/-^ (double knockout, DOKO) male mice were housed in a temperature-controlled (22 °C) facility with a strict 12 h light/dark cycle. The mice had free access to water before and during experiments and normal chow before the experiment. The diet contained 18.5% protein, 4% fat and 55.7% carbohydrate (R36 diet, Lactamin AB, Stockholm, Sweden). LXRα and β-deficient mice and corresponding controls had mixed genetic backgrounds based on 129/Sv and C57BL/6J strains and were backcrossed in C57BL/6J mice (B & K Universal Ltd, Sollentuna, Sweden) for six generations. The generation of the LXRα^-/-^, LXRβ^-/-^ and the DOKO mice have been described previously [47,48].

Male mice aged 8–12 weeks (weight 25–30 g) were fasted for 24 h (Fasted) or fasted for 24 h and refed for 12 h (Refed). All male littermates in a litter of correct age, weight and genotype were randomly allocated to a given treatment. No particular measures were taken to minimize subjective bias during group allocation. Each genotype-treatment group consisted of 5–8 animals, 53 mice in total, allowing us to detect 2-fold changes, given a coefficient of variation of 25% or less [49]. The mice were euthanized by cervical dislocation at the end of the dark period. Livers were dissected and rapidly frozen in liquid nitrogen and stored at -80 °C until isolation of total RNA. The liver samples were coded and processed by two independent, blinded researchers during RNA sample preparation and quantitative real-time polymerase chain reaction (qRT-PCR). All animal use was approved and registered by the Norwegian Animal Research authority and the regional ethical committee for animal experiments in Sweden.

### 2.4. Mouse Primary Hepatocytes Isolation and Culture 

Mouse primary hepatocytes were isolated as previously described with modest changes [50]. Briefly, male C57BL/6N mice (Jackson Laboratory, Bar Harbor, ME, USA) aged 7–8 weeks were anaesthetized with isoflurane (AbbVie, North Chicago, IL, USA), before the livers were perfused via the portal vein with liver perfusion medium (#17701038, Thermo Fisher Scientific, Bleiswijk, The Netherlands) for 15 min (2 mL/min) followed by liver digestion medium (#17703034, Thermo Fisher Scientific, Bleiswijk, The Netherlands) for 15 min. The liver was then removed and dissociated in liver perfusion buffer before filtering through a 100 μM strainer (Thermo Fisher Scientific, Bleiswijk, The Netherlands). Hepatocytes were washed 4 times with ice-cold low glucose DMEM (D6046; Sigma-Aldrich, St. Louis, MO, USA) supplemented with 10 mM Hepes (#15630080, Thermo Fisher Scientific, Bleiswijk, The Netherlands), 5% charcoal stripped FBS (#12329782, Thermo Fisher Scientific, Bleiswijk, The Netherlands) and 1% penicillin-streptomycin (50 U/mL; 50 μg/mL). Hepatocytes were seeded at 2.5 × 10^5^ cells/well onto type I collagen coated 12-well plates in attachment medium (William’s E media, #12551032, Thermo Fisher Scientific, Bleiswijk, The Netherlands) with 10% FBS, 1% penicillin-streptomycin and 10 nM insulin. The medium was changed 2 h after plating to overnight media consisting of low glucose DMEM (D6046; Sigma-Aldrich, St. Louis, MO, USA), 5% FBS, 1% penicillin-streptomycin and 1 nM insulin. On the next day, hepatocytes were washed twice with 1 mL medium/well consisting of DMEM with either 1 mM glucose (LG) or 25 mM glucose (HG), 5% FBS, 1% penicillin-streptomycin and 1 nM insulin and then cultured overnight in LG or HG medium supplemented with 0.1% DMSO, 10 µM GW3965 or 10 µM T0901317, respectively. Hepatocytes were harvested after 18 h treatment for RNA isolation.

### 2.5. Cell Culture and Transfection

Huh7 human liver hepatoma cells [51] and COS-1 fibroblast-like cells derived from African green monkey kidney (ATCC, CRL-1650™) were maintained in 25 mM glucose DMEM (D6546; Sigma-Aldrich, St. Louis, MO, USA) supplemented with 10% fetal bovine serum, 4 mM L-Gln and 1% penicillin/streptomycin. AML12 mouse liver non-cancerous cells (ATCC, CRL-2254™) were maintained in DMEM/F12 (#31331, Gibco, Thermo Fisher Scientific, Bleiswijk, The Netherlands) supplemented with 10% fetal bovine serum, 1% penicillin/streptomycin, 0.1% insulin, transferrin, and sodium selenite (ITS) (I3146; Sigma-Aldrich, St. Louis, MO, USA) and 0.1 µM dexamethasone (D2915; Sigma-Aldrich, St. Louis, MO, USA). All cells were maintained at 37 °C in humidified atmosphere of 5% CO_2_ in air and routinely tested for mycoplasma contamination. Cells were transfected with indicated plasmids using Lipofectamine 2000 (#10696153, Thermo Fisher Scientific, Bleiswijk, The Netherlands).

### 2.6. Plasmids

The FLAG-tagged human or mouse LXR expressing plasmids pcDNA3-FLAG-hLXRα, pcDNA3-FLAG-hLXRβ and pcDNA3-FLAG-mLXRα, the untagged human RXRα expressing plasmid pcDNA3-hRXRα and the empty vector pcDNA3-FLAG have been described earlier [18,52]. To generate the pcDNA3-FLAG-hLXRα-DBD-mutant, two cysteine to alanine point mutations (C115A/C118A) in the DNA binding domain were introduced using the QuikChange Site-directed Mutagenesis kit (Agilent Genomics, Santa Clara, CA, USA) and the primers listed in Appendix A. The plasmids expressing mouse Mlxγ (pCMV4-HA-mMlxγ), ChREBPα (pCMV4-FLAG-mChREBPα) and ChREBPβ (pCMV4-FLAG-mChREBPβ), as well as the empty vector pCMV4, were received as generous gifts from Prof. Mark Herman [37]. To generate the ChREBPα LID expression plasmid, cDNA corresponding to the FLAG-tag, ChRBEPα amino acids 1–178 and a stop codon, was PCR amplified from pCMV4-FLAG-mChREBPα using specific primers (Appendix A), and subcloned into the pCMV4 vector using BglII and HindIII restriction enzymes. The plasmid expressing the ChREBP quadruple mutant (H51A/S56D/ F90A/N278A) (ChREBP-Q) was a kind gift from Prof. Em. Howard Towle [32]. Accession numbers of the DNA sequences are listed as below: human LXRα, NM_005693.4; human LXRβ, NM_007121.7; mouse LXRα, NM_013839.4; human RXRα, NM_002957.6; mouse ChREBPα, NM_021455.5; mouse Mlxγ, NM_011550.3.

The *Chrebpβ* promoter-driven luciferase reporter pGL3b-ChREBPβ-exon1b-luc (wild-type) and its mutants ChoRE+Ebox-del (both ChoRE and E-box deleted) and ChoRE-del (ChoRE deleted) were kind gifts from Prof. Mark Herman [37]. The pGL3b-ChREBPβ-exon1b-luc reporter mutants Ebox-del (E-box deleted) and DR4-del (candidate DR4 response element deleted) were generated using the QuikChange Site-directed Mutagenesis kit (Agilent Genomics, Santa Clara, CA, USA) and the primers listed in Appendix A. The pGL3-rL-PK(-183)-luc (PK-luc) was a kind gift from Prof. Em. Howard Towle [53] and the ChoRE mutated reporter pGL3-rL-PK(-183)-Gal4-luc (PK-ChoREmut-luc) was a kind gift from Prof. Donald K. Scott [54]. The mouse SREBP-1c reporter pGL2basic/-550-mSREBP1c-prom-luc (SREBP1c-luc) was kindly provided by Prof. Nobuhiro Yamada [55].

The multimerized ChoRE and LXRE reporters: pGL3b-2xChoRE-2xLXRE-10 (ChoRE+LXRE-luc), pGL3b-2xChoRE-10 (ChoRE-only-luc) and pGL3b-2xLXRE-10 (LXRE-only-luc) were constructed in two steps—first, the 136 bp inserts were made by gene synthesis (GenScript, Nanjing, China). The SacI-BglII-lined inserts were designed to contain two canonical ChoREs (CACGTGatataCACGTG) and two canonical LXREs (AGGTCActctAGGTCA) in the order ChoRE-LXRE-ChoRE-LXRE and with a spacing of 10 bp (gtaatattaa), giving a phasing of ~20 bp center to center between the REs. The inserts were then cut from the production vector pUC57 (GenScript, Nanjing, China) and subcloned into pGL3 basic vector between SacI and BglII. The schematic representation of synthetic reporters is illustrated in Figure 3A. The Renilla Luciferase reporter pRL-CMV (Promega, Madison, WI, USA) was used as internal control of transfection efficiency. All plasmids were verified by sequencing.

### 2.7. Luciferase Reporter Assay

Huh7 cells were seeded at density 7 × 10^4^ cells/well in 24-well plates. After 24 h, cells were transfected with 300 ng of luciferase reporters, 150 ng of LXR/RXR and/or 150 ng of ChREBP/Mlx expressing plasmids or empty controls pcDNA3-FLAG and pCMV4. Renilla luciferase reporter (pRL-CMV; 50 ng) was included as an internal control for transfection efficiency. All transfections were performed with Lipofectamine 2000. Dual luciferase reporter assay was performed 24 h post transfection as previously described [16]. Transfections were verified by immunoblotting. For LXR agonist treatment, synthetic ligand GW3965 (1 µM or 10 µM) or T0901317 (5 µM) was applied to cells 6 h post transfection and 0.1% DMSO was used as control. For low vs high glucose (2.5 mM vs 25 mM) treatment, cells were transfected in high glucose and 6 h post transfection cells were washed twice with low or high glucose medium, respectively and then cultured in corresponding medium for 18 h. After 18 h of incubation, cells were washed with PBS and lysed in Passive Lysis Buffer (Promega, Madison, WI, USA). Dual-Luciferase® Reporter Assays (Promega, #E1960) were run on a Synergy H1 plate reader (BioTek® Instruments, Winooski, VT, USA) according to the manufacturer’s manual. Readings of Firefly Luciferase were normalized to the Renilla Luciferase readings and data from at least three independent transfections experiments run in quadruplicates are presented.

### 2.8. Chromatin Immuneprecipitation (ChIP)

Huh7 cells were transfected with the ChoRE+LXRE reporter and plasmids expressing ChREBPα, ChREBP-Q or ChREBPβ and Mlxγ. AML12 cells were transfected with plasmids expressing ChREBPα, Mlxγ, with or without LXRα and RXRα. For LXR agonist treatment, 10 µM of GW3965 was applied to cells 6 h post transfection and 0.1% DMSO was used as control. Twenty-four hours post transfection, cells were cross-linked with 1% formaldehyde for 10 min at room temperature, followed by 5 min incubation with 125 mM glycine to quench the reaction. Cells were washed twice in cold PBS and harvested in PBS-T. The cell pellets were lysed in lysis buffer (0.1% SDS, 1% Triton X-100, 0.15 M NaCl, 1 mM Ethylenediaminetetraacetic acid (EDTA) and 20 mM Tris (pH 8.0). The lysed cells were sonicated to get an average size of 200–500 bp chromatin fragments using a Bioruptor (Diagenode, Seraing, Belgium). Chromatin was immunoprecipitated with 4 µg antibody against ChREBP (NB400-135; Novus Biologicals Centennial, CO, USA), against LXR (antibody generation described in Reference [56]), or rabbit IgG (011-000-002; Jackson ImmunoResearch Laboratories, West Grove, PA, USA) over night at 4 °C. Protein A Dynabeads were washed four times in lysis buffer, added to the chromatin and rotated at 4 °C for 2 h. The Dynabeads were then washed three times with wash buffer 1 (0.1% SDS, 1% Triton X-100, 0.15 M NaCl, 1 mM EDTA, 20 mM Tris (pH 8)), followed by washing once in wash buffer 2 (0.1% SDS, 1% Triton X-100, 0.5 M NaCl, 1 mM EDTA, 20 mM Tris (pH 8)), once in wash buffer 3 (0.25 M LiCl, 1% NaDOC, 1% NP-40, 1 mM EDTA, 20 mM Tris (pH 8)) and finally once in wash buffer 1. All washing steps were done with rotation for 5 min at room temperature. DNA-protein complexes were eluted with 1% SDS and reverse cross-linked over night at 65 °C. DNA was purified by using the QIAquick PCR Purification Kit (#28104; QIAGEN, Hilden, Germany). DNA enrichment was quantified by qRT-PCR. ChIP primer sequences are listed in Appendix A.

### 2.9. RNA Extraction, cDNA Synthesis and Quantitative RT-PCR

RNA was isolated with TRIzol® reagent (#15596018, Invitrogen, Thermo Fisher Scientific, Waltham, MA, USA) according to the manufacturer’s protocol, including a high salt (0.8 M sodium acetate, 1.2 M NaCl) precipitation step to avoid contaminating polysaccharides to co-precipitate with RNA. Extracted RNA was further purified using RNeasy spin columns (#74104, QIAGEN, Hilden, Germany). Isolated RNA (500 ng) was reverse transcribed into cDNA using MultiScribe Reverse Transcriptase (#4311235, Thermo Fisher Scientific, Bleiswijk, The Netherlands) and random hexamer primers. RT-qPCR was performed with 1 µL of the cDNA synthesis reaction using Kapa SYBR FAST qPCR Master Mix (KapaBiosystems, Roche, Basel, Switzerland) on a Bio-Rad CFX96 Touch™ Real-Time PCR Detection System (Bio-Rad Laboratories, Hercules, CA, USA). Assay primers were designed with Primer-BLAST software (NCBI, Bethesda, MD, USA) [57]. Gene expression was calculated using the 2^-ΔΔCT^ method and normalized against the expression of TATA-binding protein (*Tbp*). All primer pairs showed an efficiency of 90–110% at R^2^ > 0.98. Primer sequences are listed in Appendix A.

### 2.10. Co-Immunoprecipitation (CoIP)

COS-1 cells were transfected with plasmids expressing ChREBPα full-length (FL) or N-terminal LID truncation or ChREBPβ, with or without LXRα FL or truncations (amino acids: 1–166, 95–447 and 166–447) or LXRβ. For ChREBPβ transfections we used 6-fold more DNA than of ChREBPα to obtain comparable protein levels in the CoIP. For LXR agonist treatment, 0.1% DMSO or GW3965 (1 µM) was applied to cells 6 h post transfection. Cells were harvested 24 h post transfection and lysed in lysis buffer (200 mM NaCl, 20 mM HEPES (pH 7.4), 1% NP-40) containing 1 mM NaF, 1 mM Na_3_VO_4_, 1 mM β-glycerophosphate and Complete^TM^ protease inhibitors (Roche Applied Science, Penzberg, Germany), followed by snap freezing on dry ice to disrupt nuclear membranes. The lysates were then thawed on ice, cleared by centrifugation at 17,000× *g*, 10 min and immunoprecipitated with 2 μg ChREBP (NB400-135, Novus Biologicals, Littleton, CO, USA), LXRα (PP-PPZ0412; R&D Systems, Minneapolis, MN, USA) or LXRβ (PP-K8917; R&D Systems, Minneapolis, MN, USA) antibodies bound to protein A Dynabeads (Invitrogen, Thermo Fisher Scientific, Waltham, MA, USA) for 2 h at 4 °C. Beads were washed three times in wash buffer (200 mM NaCl, 20 mM HEPES (pH 7.4), 0.1% NP-40) with 5 min rotation in between, before proteins were eluted from the beads with 1× SDS loading buffer at 95 °C for 5 min. Co-immunoprecipitated proteins were analyzed by immunoblotting.

### 2.11. Immunoblotting

Proteins were separated by Sodium dodecyl sulfate polyacrylamide gel electrophoresis (SDS-PAGE) (Bio-Rad, Hercules, CA, USA) and blotted onto PVDF membrane (MerckMillipore, Darmstadt, Germany). Primary antibodies used were LXRα (PP-K8607, R&D systems; 1:1000), LXRα LBD (PP-PPZ0412-00, R&D systems, Minneapolis, MN, USA; 1:1000), LXRβ (PP-K8917-00, R&D systems, Minneapolis, MN, USA; 1:1000), RXRα (sc-553, Santa Cruz Biotechnology, Dallas, TX, USA; 1:1000), ChREBP (NB400-135, Novus Biologicals, Littleton, CO, USA; 1:1000), FLAG (F1804; Sigma-Aldrich, St. Louis, MO, USA; 1:1000) and β-actin (A5441; Sigma-Aldrich, St. Louis, MO, USA; 1:1000). Secondary antibodies used were horseradish peroxidase-conjugated goat anti-mouse IgG or mouse anti-rabbit IgG (115-035-174 and 211-032-171, Jackson ImmunoResearch Laboratories, West Grove, PA, USA; both 1:10,000). 

### 2.12. Statistical Analysis

Statistical analyses were performed using GraphPad Prims 8 (GraphPad Software Inc., San Diego, CA, USA). All data were presented as means and standard error of the mean (SEM). In addition, all individual data points are plotted. Data distribution and similarity in variance between groups were analyzed by Shapiro-Wilk and Brown-Forsythe tests, respectively. Statistical differences between groups were determined by two-way analysis of variance (ANOVA) followed by Tukey’s multiple comparison tests for data with two variables. For data with one variable we used one-way ANOVA with Dunnett correction to compare every mean to the control mean, alternatively Tukey correction for multiple comparisons. For all statistical tests, *p* < 0.05 was considered statistically significant.

### 2.13. Gene Set Enrichment Analysis 

Gene set enrichment analysis of genes with ChREBP-LXR peak pairs with a peak-to-peak-distance <100 bp, residing in promoters, TSS and first exons (based on the *Rgmatch*; corresponding to −5000 to +2500 bp from TSS) were made with the ConsensusPathDB tool from the Max-Planck-Institute for Molecular Genetics (http://cpdb.molgen.mpg.de/) [58], using the Reactome Pathway Database (https://reactome.org). Minimum overlap with input list was set to 3 and the *p*-value cutoff to *p* < 0.01.

### 2.14. Data Availability

The ChIP-seq datasets used in this study is available at NCBI GEO (https://ncbi.nlm.nih.gov/geo) under accession number GSM864670 (LXR), GSM1899651 (FXR) and GSM864671 (PPARα) without any restrictions. The ChREBP ChIP-seq dataset [42] was kindly provided by Prof. Lawrence Chan. Scripts used to perform peak calling and analyze the called peaks are available at https://github.com/ivargr/open-the-lid-chip-seq-analysis. The Forbes co-occurrence analysis is available at https://hyperbrowser.uio.no/hb/u/ivar/h/open-the-lid-co-occurence-analysis.

## 3. Results

### 3.1. ChREBPα and LXRα Interact and Show a High Co-Occupancy of the Mouse Liver Genome

LXR is one of the factors that regulate ChREBP-dependent transcription in liver. We previously demonstrated that when knocking out LXR in mice, ChREBP loses its ability to bind to carbohydrate responsive elements (ChoREs), which affects hepatic gene expression of ChREBP-specific targets like *Lpk* and *Chrebpβ* [16]. This can be explained by direct and indirect mechanisms. To better understand how LXR affects ChREBP activity, we asked whether the two TFs could interact. We co-immunoprecipitated (CoIP’ed) LXR and ChREBP in COS-1 cells cultivated in high glucose media (25 mM), to induce the activity of ChREBPα [16,37]. LXRα was detected in the ChREBP immunoprecipitate and reciprocally, ChREBPα was detected when immunoprecipitating with an LXRα antibody (Figure 1A). We next assessed whether this interaction could be detected at the genomic level. To determine this, we reanalyzed two published chromatin immunoprecipitation sequencing (ChIP-seq) datasets; one ChREBP ChIP-seq performed in livers from mice fasted and refed a high-carbohydrate diet [42] and one LXR ChIP-seq done in livers from mice treated with the LXR agonist T0901317 [43]. The sequence reads were mapped to the mouse genome generating a genome-wide, high-resolution map of ChREBP and LXR binding sites. We detected 48,647 and 24,728 binding sites for ChREBP and LXR, respectively.

Using the top 20,000 ChREBP peaks and all the LXR peaks we calculated the peak(ChREBP)-to-peak(LXR) distance for all peaks. Interestingly, as many as 11,022 peak pairs showed a peak-to-peak distance of less than 1000 bp and 7928 (71.9%) of these were less than 100 bp apart (Figure 1B). Given the resolution of these datasets, all peaks less than 100 bp apart are likely to represent co-localized peaks, indicating that ChREBP and LXR occupy many of the same loci and that co-occupancy within these regions is high (Figure 1D). We also found that the peak pairs clustered around transcription start sites (TSSs), with the expected depletion at TSS (Figure 1C). The peak pairs which represent peaks less than 100 bp apart were located both upstream and downstream of TSS. However, we found that peak pairs of *bona fide* LXR or ChREBP target genes (blue dots) located mainly upstream or near the TSS, consistent with the notion that most TF binding sites are found upstream of TSSs [59,60]. As an example, the two ChREBP target genes *Lpk* and *Chrebpβ* showed significant ChREBP enrichment at the expected sites in the respective promoters and LXR co-occupied the same sites (Figure 1E). Looking at the genomic position of all peak pairs, independent of distance to TSS using *Rgmatch* (https://bitbucket.org/pfurio/rgmatch), we found the peak pairs to be distributed throughout the genome like other NRs [43,61,62]. Specifically, 39% of them seem to cluster around TSSs (Figure 1F; Promoter-TSS-First Exon-First Intron). 

**Figure 1 cells-09-01214-f001:**
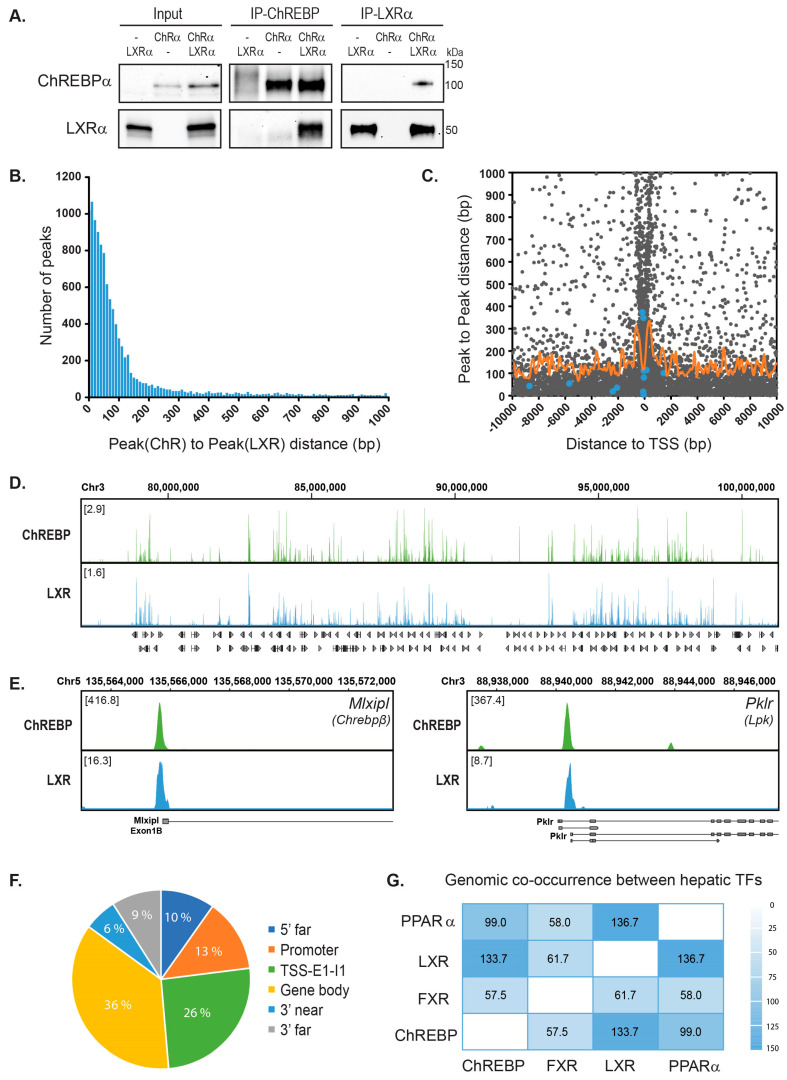
Genome-wide co-occupancy of carbohydrate responsive element-binding protein (ChREBP) and liver X receptor (LXR) in mouse liver. (**A**). Co-immunoprecipitation (CoIP) of LXRα and ChREBPα transfected in COS-1 cells cultured in 25 mM glucose. Lysates were immunoprecipitated with ChREBP and LXRα antibodies and input and immunoprecipitated proteins immunoblotted with the same antibodies (*n* = 3). One representative western blot is shown. (**B**). Distribution of ChREBP-LXR peak pairs. ChREBP ChIP-seq data from fasted and high-carbohydrate refed mouse liver [42] and LXR ChIP-seq data from T0901317-treated mouse liver [43] were reanalyzed to generate a genome-wide map of ChREBP and LXR binding sites. The top 20,000 peaks from each dataset were used to calculate the Peak(ChREBP)-to-Peak(LXR) distance and all peak pairs with a peak-to-peak distance <1000 bp were plotted against the number of peak pairs. (**C**). Localization of ChREBP-LXR peak pairs. The Peak(ChREBP)-to-Peak(LXR) distance were plotted against the distance from mid peak-to-peak position to transcription start site (TSS) of the nearest gene (Distance to TSS). Blue dots, verified LXR/ChREBP target genes, *Acaca, MlxiplA, MlxiplB, Pklr, Scd1, Srebf1* and *Txnip*. Orange curve, moving average (window: 100 bp) of peak-to-peak distance as a function of distance to TSS. (**D**). Global landscape of LXR-ChREBP co-occupancy. Browser view of LXR and ChREBP tracks on Chromosome 3. Square brackets indicate the scale maxima of log2(ChIP/input) ratios. (**E**). Local pattern of LXR-ChREBP co-occupancy. Browser view of LXR and ChREBP tracks on the promoter regions of the ChREBP target genes *Mlxipl* (*Chrebpβ*) and *Pklr* (*Lpk*). Square brackets indicate the scale maxima of ChIP/input ratios. (**F**). Genomic positions of the ChREBP-LXR peak pairs. The peak pairs were mapped to genomic elements using *Rgmatch.* The elements shown are grouped as follows: 5’ far, 25 kb to 5 kb upstream of TSS; Promoter, <5 kb upstream of TSS; TSS, −200 bp to +200 bp; E1, first exon; I1, first intron; Gene body, the whole area of any exon or intron, other than the first exon and the first intron of the gene; 3’ near, <5 kb downstream of TSS; 3’ far, 5 kb to 25 kb downstream of TSS. (**G**). Genome-wide co-occurrence of mouse hepatic transcription factors (TFs). Comparison of LXR and ChREBP with published binding profiles of peroxisome proliferator-activated receptor-α (PPARα) [43] and farnesoid X receptor (FXR) [63]. Genomic co-occurrence (to which degree binding sites occur at the same positions in the genome) between ChREBP, LXR, FXR and PPARα were measured by the Forbes coefficients (FC) using the Genomic HyperBrowser.

To assess how the seemingly high genome-wide co-occupancy of ChREBP and LXR compared with the co-occupancy between other hepatic TFs, we reanalyzed ChIP-seq data of mouse peroxisome proliferator-activated receptor-α (PPARα) and farnesoid X receptor (FXR) [43,63]. We next calculated the Forbes coefficients (FC), evaluating genomic co-occurrence while correcting for genome coverage [64]. Both PPARα and FXR are known to co-occupy genomic loci and/or interact with LXR and ChREBP [43,65,66,67]. PPARα and LXR showed the highest similar score among all compared TFs (FC: 136.7; Figure 1G), in line with published data [43]. ChREBP and LXR shared the second highest genomic co-occurrence, with a FC of 133.7 (Figure 1G). FXR was recently reported to physically interact with ChREBP, negatively regulating ChREBP activity [65]. Thus, it was somewhat unexpected to see that FXR shared a significantly smaller portion of the genome with ChREBP than LXR did (FC: 57.5; Figure 1G).

Finally, we performed gene set enrichment analysis (GSEA) based on the Reactome Pathway Database. We analyzed genes with ChREBP-LXR peak pairs residing either in the promoter, around the TSS or in the first exon and with a peak-to-peak-distance <100 bp (corresponding to peak pairs between −5000 to +2500 bp from the TSS). As expected, the filtered genes were enriched in pathways regulated by both ChREBP and LXR, for example, metabolism of lipids, fatty acids and glucose (Table 1). Notably, pathways involved in metabolism of amino acids, γ-carboxylation and coagulation, which has not been described to be regulated by ChREBP nor LXR, were also enriched in the gene set. This suggests potential novel biological functions regulated through the concomitant binding of ChREBP and LXR.

### 3.2. LXRα and ChREBPα Co-Activates ChREBP Target Genes In Vitro and In Vivo 

Having established that ChREBPα and LXRα interact and that they co-occupy regulatory regions in the mouse liver genome, we addressed the transcriptional effects of this interaction in reporter gene assays using two luciferase reporter constructs driven by the mouse *Chrebpβ* promoter [37] and the rat *Lpk* promoter [53]. LXR was always co-transfected with its dimerization partner RXRα and ChREBP with its dimerization partner Mlxγ, unless otherwise stated. As expected, ChREBPα:Mlxγ were able to induce the transcription from both reporters in the human hepatocarcinoma cell line Huh7 (Figure 2A). Moreover, ChREBPα transactivation of the *Chrebpβ* promoter increased three-fold when the glucose concentration in the media was augmented from 2.5 mM to 25 mM (Figure 2A). This is consistent with data from the seminal ChREBPβ study by Herman and coworkers [37]. Interestingly, LXRα:RXRα seemed to be able to upregulate the transcriptional activity from both promoters and ChREBPα and LXRα together showed a synergistic effect on the transcriptional activity (Figure 2A). To exclude the potential impact of RXRα in regulating these constructs, we examined RXR and LXR separately and observed no induction of *Chrebpβ* promoter activity from either RXRα or LXRα alone (Appendix A). In the following in vitro experiments, we chose to keep the glucose concentration at 25 mM to ensure a high ChREBPα activity, unless otherwise stated.

To investigate the impact of the different LXR subtypes, LXRα and LXRβ, on ChREBP activity in vivo, we performed fasting-refeeding experiments with wild-type (WT), LXRα^-/-^, LXRβ^-/-^ and LXRα^-/-^β^-/-^ (double knockout, DOKO) mice. The mice were fasted for 24 h or fasted for 24 h and refed for 12 h before they were sacrificed. Liver from 5–8 animal per genotype were examined. The feeding-induced hepatic expression of the ChREBP specific target genes *Lpk* and *Chrebpβ* was significantly reduced in LXRα^-/-^ and DOKO mice but not in LXRβ^-/-^ mice (Figure 2B). On the contrary, no significant changes in *Chrebpα* expression was observed that could explain the different responses to LXRα versus -β knockout seen for the ChREBP target genes, suggesting that LXRα but not LXRβ, is essential to regulate ChREBP activity but not *Chrebpα* expression. This conclusion was substantiated in reporter gene assays, showing no activation of the *Chrebpβ* promoter by LXRβ and no synergistic effect when co-expressing LXRβ and ChREBPα (Appendix A). Taken together, these results demonstrate that LXRα but not LXRβ, co-activates the expression of ChREBP-specific target genes in vitro and in vivo together with ChREBPα.

### 3.3. LXRα:ChREBPα Co-Activation Requires Functional ChoREs But Not LXREs

LXR and ChREBP share many transcriptional targets, for example, the DNL genes *Acc*, *Fasn*, and *Scd1*, which contain both LXREs and ChoREs in their promoter/regulatory regions [3,21]. The ChREBP target genes *Lpk* and *Chrebpβ* promoters, however, only contain ChoREs and no canonical LXREs. Analysis of the *Lpk* and *Chrebpβ* reporters using *NHR scan* (http://www.cisreg.ca/cgi-bin/NHR-scan/nhr_scan.cgi) suggested one weak candidate in the *Chrebpβ* exon 1B promoter. This was, however, discarded after a deletion scan showing no effect on the promoter activity (Appendix A; DR4 del). We also remapped the functional ChoRE in the *Chrebpβ* exon 1B promoter to the so-called “E-box-like” domain, 98 bases upstream of the TSS (Appendix A)—a detail missed in the original ChREBPβ paper [37]. Nonetheless, despite no obvious LXRE, LXRα was able to activate the reporters together with ChREBPα (Figure 2A). We therefore investigated the impact of ChoREs and LXREs on LXRα:ChREBPα co-activation by designing three synthetic luciferase reporters: one containing two canonical ChoREs and two canonical LXREs, one that contained only the two ChoREs and one that only contained the two LXREs. The order (ChoRE-LXRE-ChoRE-LXRE) and the phasing (20 bp) of the recognition elements was kept constant (Figure 3A) to make sure that the TFs would adopt the same rotational orientation in all three constructs, as this has been shown to greatly impact transactivity on compound promoters [68]. To ensure an acceptable read-out from the synthetic promoters, we turned to the constitutively active forms of ChREBP, ChREBPβ and ChREBP-Q. While ChREBPβ is a naturally occurring isoform lacking the LID, ChREBP-Q is a ChREBPα quadruple mutant H51A/S56D/F90A/N278A that has escaped the low-glucose inhibition while keeping the N-terminal domain [32]. As expected, a synergistic co-activation was observed with both LXRα/ChREBP-Q and LXRα/ChREBPβ on the ChoRE+LXRE reporter (Figure 3B), reflecting the situation on a dual LXR/ChREBP target gene. Interestingly, this pattern was retained on the ChoRE-only reporter but only with LXRα and ChREBP-Q. ChREBPβ seemed to have lost the ability to co-activate with LXRα in this context, mimicking a ChREBP-specific target gene (Figure 3B). This was not due to different abilities to bind to the promoter, as evaluated by ChIP (Appendix A). The effect was even more dramatic on the LXRE-only reporter where neither ChREBP-Q nor ChREBPβ was able to activate, even though LXR still transactivated the promoter (Appendix A). This indicates that ChoREs but not LXREs, are sufficient to support LXRα:ChREBPα co-activation and that only full-length ChREBP, that is, not ChREBPβ, is co-activated by LXRα on ChoRE-only target genes. To assess this on natural promoters we used the ChREBP-specific, ChoRE-only *Chrebpβ* and *Lpk*-driven reporters. Again, the same pattern emerged: LXRα was able to co-activate ChREBP-Q but not ChREBPβ that lacks most of the LID (Figure 3C).

### 3.4. LXRα and ChREBPα Interact via Key Activation Domains

The ability of LXRα to co-activate ChREBPα in an LXRE-independent manner naturally brought up the question of what domains of LXRα and ChREBPα that are involved in their interaction. We therefore examined the interaction between full-length LXR and ChREBP and different truncations using CoIP. ChREBPα interacted with LXRα when immunoprecipitating both ways (Figure 1A and Figure 4A). On the other hand and in line with the gene expression data (Figure 3), ChREBPβ did not bind to LXRα as neither ChREBPβ nor LXRα was able to co-immunoprecipitate with each other (Figure 4A). ChREBPβ lacks the first 177 amino acids (aa) in the N-terminal, which form most of the LID in ChREBPα. Hence, we constructed a ChREBP truncation that only expressed the first 177 residues, that is, the LID. This domain bound strongly to LXRα and seemed sufficient to support the interaction between the full-length factors (Figure 4A). Furthermore, we examined the interactions among ChREBPα and different LXRα truncations. ChREBPα interacted with the LXRα C-terminus, including its hinge domain and LBD (Figure 4B). In addition, ChREBPα also interacted with LXRβ (Appendix A), although this interaction seems to be less relevant in liver (Figure 2B and Appendix A). ChREBPα was previously shown to interact with FXR, with one of the interaction surfaces residing in its LBD [65]. Multiple sequence alignment using ClustalX (http://www.clustal.org/clustal2) revealed that LXRα, LXRβ and FXR show high similarity in multiple patches throughout the LBD (data not shown), suggesting that LXRα might use a similar interface when binding to ChREBPα.

To expand on the consequences of the ChREBP:LXR interaction in a functional context, we asked whether LXRα is dependent on binding to DNA to induce the activity of ChREBPα on ChREBP target genes, given that the interaction seems to run via LID and LBD. To this end, we constructed an LXRα DNA binding mutant (DBDm), in which two cysteines (C115 and C118) in the first DNA-binding domain (DBD) zinc finger were mutated to alanine to abrogate DNA binding. The LXRα DBD mutant had lost its ability to transactivate the *Srebp1c* promoter as well as the *Lpk* promoter (Appendix A), the former being a well-established LXR target gene [55]. Peculiarly, the LXRα DBDm also lacked the ability to co-activate ChREBPα, arguing that DNA binding or rather DBD integrity is important for the full co-activation effect of LXR (Appendix A). This is in line with what we observed in Figure 3B, where the promoter LXRE was mutated and the reporter output amplitude dropped while the regulation pattern was retained. Moreover, it contrasts with the loss of co-regulation seen when the ChoRE was mutated (Appendix A, right panel; Figure 3B, right panel). Taken together, our data suggest that ChREBPα and LXRα interact via key regulatory domains, namely the N-terminal LID of ChREBPα and the C-terminal, LBD-containing part of LXRα. This assigns a novel function to the ChREBP LID in physically bridging the two factors, allowing LXRα to co-activate ChREBPα on ChREBP-specific target genes. Importantly however, our data do not formally exclude the possibility of other factors, like RXR or Mlx, being involved in tethering the two domains.

### 3.5. Ligand-Activated LXRα Represses ChREBPα Activity on ChREBP-Specific Target Genes

The physical interaction and co-regulatory interplay between ChREBPα and LXRα led us to ask how this affects gene expression in a fully chromatinized context. To investigate this, we isolated primary mouse hepatocytes and cultivated them for 24 h in either low glucose (1 mM) or high glucose (25 mM) to induce ChREBP activity and concomitantly stimulated them with the potent, selective LXR agonist GW3965 (10 μM) to induce LXR-activity. The DNL genes *Acacb*, *Fasn* and *Scd1*—which are common targets of LXR and ChREBP—were significantly upregulated by the LXR agonist (Figure 5A). The expression of ChREBP-specific target genes *Chrebpβ, Lpk*, *Txnip* and *Rgs16* on the other hand showed a dramatically different pattern (Figure 5B): These genes were upregulated by high glucose treatment, while the LXR agonist surprisingly displayed a repressive effect under high glucose conditions. This was also seen with the *Chrebpβ* reporter (Appendix A). To exclude a possible GW3965 peculiarity, we recapitulated this experiment with the same outcome using Tularik (T0901317), another LXR agonist, both in primary hepatocytes and by transfecting the ChREBP-specific *Lpk* reporter in Huh7 cells (Appendix A). These data reveal that ligand-activated LXRα plays distinct roles on different groups of genes, activating common targets of LXR and ChREBP, while repressing ChREBP-specific target genes.

### 3.6. Ligand-Activated LXRα Reduces ChREBP Binding to Chromatin

To try to untangle the mechanism underlying the LXR ligand-dependent repressive effect, we performed ChIP assays to study ChREBPα and LXRα chromatin binding dynamics in the non-cancerous, mouse hepatocyte cell line AML12. Based on our analysis of the ChIP-seq datasets (Figure 1), we selected four gene loci that are co-occupied by ChREBP and LXR in the promoter region: the ChREBP-specific targets *Lpk* and *Txnip* and the common LXR and ChREBP targets *Fasn* and *Scd1* (Figure 6A). The *Mlxipl* exon1b promoter was excluded from these analyses due to low expression of ChREBPβ in AML12 cells (data not shown). In the presence of ChREBPα, all four promoters were robustly immunoprecipitated. Moreover, LXRα did not affect ChREBPα occupancy on any of the promoters (Appendix A). Conversely, ChREBPα increased the binding of LXRα on the same promoters (Appendix A), in line with the notion that ChREBPα is able to recruit LXRα to ChoREs via the LID-LBD interaction (Figure 3 and Figure 4).

When the same cells, transfected with both ChREBPα and LXRα, were treated with LXR agonist GW3965, ChREBPα-binding to chromatin was reduced on all four promoters (Figure 6A). Concomitantly, a weak, non-significant reduction was observed for LXRα occupancy. Supporting these observations, GW3965 treatment reduced the ChREBPα:LXRα interaction (Figure 6B), suggesting that a ligand-induced conformational change in LXRα LBD reduces its affinity for ChREBPα. Loss of LXR from the complex weakens ChREBPα’s ability to bind chromatin. This likely has the most dramatic effect on ChREBP-specific target genes, like *Lpk* and *Txnip*, where the LXR binding to the promoter chromatin is mediated through ChREBP (Figure 6A left panel). Common LXR and ChREBP target gene promoters, like *Fasn* and *Scd1*, should still be able to accommodate both factors (Figure 6A). Altogether, this results in the reduced expression of ChREBP-specific targets genes seen in Figure 4B. This echoes the lost ChoRE binding and lost feeding-induced expression of ChREBP-specific target genes we previously reported with the LXR double knockout mice [16]. The current data broaden this picture, showing that LXR ligand engagement modulates ChREBPα:LXRα interaction, chromatin occupancy and target gene co-activation.

## 4. Discussion

Altogether, our findings argue for a close collaboration between LXRα and ChREBPα in regulating glycolytic and lipogenic genes. Hence we propose a new model (Figure 7): Upon glucose signals, the physical interaction between LXRα and ChREBPα allows unliganded LXRα to be recruited to ChoRE-containing promoters and increase ChREBP-specific target gene expression; upon agonist/oxysterol signals, liganded LXRα functions as a molecular switch, disassembling the LXRα:ChREBPα complex and ensuring an adequate restriction of glycolytic and lipogenic target genes.

The transcriptional regulation of ChREBP expression by LXR was determined by Cha and Repa already in 2007 [19]. Since then the idea has been that LXR, in addition to regulating its own direct target genes, indirectly contributes to the induction of ChREBP regulated glycolytic and lipogenic target genes. However, in this study we show that the picture is more complex. In addition to upregulating *Chrebpα* expression by binding to LXREs and activating the *Mlxipl* promoter, LXRα can directly regulate the transcription of ChREBPα target genes by binding to ChREBPα. With this we propose a new mechanism of action for LXR: *trans-coactivation* (Figure 7). Trans-coactivation resembles transrepression observed with several nuclear receptors (NRs) including GR, PPARγ, FXR and LXRs [65,69,70,71], as both mechanisms rely on the NR being tethered to other TFs *in trans*. However, while LXR transrepression of NFκB and AP-1 is agonist-dependent [71,72], LXR trans-coactivation of ChREBP relies on unliganded LXR (Figure 7). Our findings indicate that LXR might have DNA-binding domain (DBD)-independent function that contributes to its regulation of liver metabolism. This was recently also demonstrated for Rev-erbα, which is tethered to chromatin by hepatic lineage determining TFs [73]. DBD-independent functions like these might explain non-overlapping cistromes and/or transcriptomes in different cell types [43,74]. A relocation of LXR from trans-coactivation complexes on ChREBP-specific target genes to classical ligand-engaged RXR heterodimers on LXR-specific target genes is also in line with the type of TF-cofactor redistribution described for NFκB [75] and PPARγ [76]. An LXR-driven redistribution of co-factors between ChREBP-specific and LXR-specific promoters, would also explain the ligand-dependent repression observed in our study. In fact, SREBP-1c could be one of the potential candidates, as LXR, ChREBP and SREBP-1c are tightly interconnected and coordinately regulate lipogenesis. SREBP-1c is known to compete with PGC1α for direct interaction with the LBD of HNF-4 [77,78]. Ligand-activated LXR may lead to increased levels of SREBP-1c proteins that potentially could compete with the interaction between LXR and ChREBP.

Posttranslational modification by small ubiquitin-like modifier (SUMO) of TFs and coregulators is generally linked to transcriptional repression [79] and SUMOylation of LXRs and PPARγ appears to be required for transrepression in macrophages [71]. Phosphorylation and acetylation, which reflect intracellular nutrient availability, modulate ChREBP and LXR transactivity, co-factor recruitment, DNA binding and stability [80,81]. Whether these posttranslational modifications are involved in enforcing the LXRα:ChREBPα complex is not known. If so, O-GlcNAc modification, which derives from the metabolically integrated hexosamine biosynthetic pathway (HBP), might be a candidate. LXR is post-translationally O-GlcNAc modified in response to high glucose [16,18]. The same is true for ChREBP, which leads to increased transcriptional activity and recruitment to target gene promoters [38,39]. Importantly, LXR and the O-GlcNAc transferase (OGT) bind directly to each other and co-localize in the nucleus. Moreover, LXR regulates the O-GlcNAc-modification of ChREBPα [16]. Could LXRα be involved in tethering OGT to ChREBPα and through this mediate O-GlcNAc modification, as reported by Guinez et al. [39] and subsequently cause the synergistic trans-coactivation of ChREBPα target genes shown herein? This is an interesting possibility that would link the nutrient sensor O-GlcNAc and the flux through the HBP even tighter to the cross-regulation of LXR and ChREBP [4,16,40,82].

An important observation in our study is the different ability of ChREBPα and ChREBPβ to respond to the LXRα trans-coactivation. While ChREBPα and ChREBP-Q are activated by LXRα and repressed by LXR agonists, ChREBPβ is non-responsive to both cues (Figure 2, Figure 3 and Appendix A). This is due to ChREBPβ lacking a functional LID, which directs the interaction with LXR. The LID, located in the ChREBPα N-terminal—also called the MondoA conserved region (MCR)—was originally described as a purely repressive domain [31,83]. However, by mutating the ChREBPα N-terminal Davies et al. showed that this domain is indeed an activation domain kept silent at low glucose conditions [32]. Moreover, they speculated that the MCR1–4 domains (ChREBP aa 1–197) must be interacting with a coregulatory protein that plays a role in transcriptional activation in a step subsequent to glucose-dependent relief of repression [32]. In a bioinformatics sequence-structure analysis approach ChREBP was in fact suggested to interact with NRs, like HNF4α and FXR but through a nuclear receptor box (NRB) in the proline-rich region (Figure 4A), to support interaction between CBP/p300 and ChREBP MCR6 [33]. Based on our data, we now propose LXRα to be the co-regulator hypothesized by Davies et al. [32]. Moreover, we confirm that ChREBPβ has escaped low-glucose control and we expand this to include negative regulation by liganded LXR. As a consequence ChREBPβ acts as an extremely potent effector of carbohydrate signals through its feed-forward relationship with ChREBPα and so it is no surprise that its expression levels and stability is much lower than ChREBPα. This has been described before [37,84] and is confirmed in our study (Figure 4).

The high genome-wide co-occupancy of LXR and ChREBP observed in the ChIP-seq data was rather surprising: More than 71% of ChREBP-LXR peak pairs have peaks less than 100 bp apart (Figure 1B), suggesting full co-localization at these sites. A similar frequency of overlapping LXR peaks had been reported before by Boergesen et al. for LXR and PPAR, on non-canonical LXR/PPAR recognition elements (DR4 or DR1) [43]. The existence of LXR-PPARα heterodimers has been suggested earlier [67] but the authors found no evidence for such transcriptional complexes. Instead they concluded that the receptors bound to the same degenerate elements, representing NR binding hot spots [85], in a mutually exclusive manner [43]. In our study, on the other hand, many of the overlapping LXR and ChREBP peaks are found on or adjacent to ChoREs (in e.g., *Lpk, Chrebpβ, Tixnip, Rgs16*), with no obvious LXRE/DR4 close by. In addition, our data demonstrate a physical interaction between the factors and the ability of ChREBPα to recruit LXRα to the promoter ChoRE but not the other way around. This points to a different mechanism underlying the ChREBP-LXR co-occupancy, where ChREBPα recruits LXRα in trans to ChoREs. Our data do not formally rule out the possibility that LXRα on certain sites also contacts chromatin directly, via its DBD, in addition to binding to ChREBPα via its LBD. In fact, such ChREBP-LXR contacts might function as bridging point for smaller or larger chromatin loops [86,87], like promoter-enhancer loops [88], possibly within topologically associated domains (TADs) [89]. A tantalizing scenario could be that ChREBPα and LXRα are involved in bridging the exon 1A (*Chrebpα*) and exon 1B (*Chrebpβ*) promoter at the *Mlxipl* locus, where ChREBPα binding to the *Chrebpβ* promoter could contact LXRα sitting on the *Chrebpα* promoter 17 kb downstream and coordinate the expression from both promoters given different nutritional cues. High glucose would enforce the contact, driving *Chrebpβ* expression [37], while high levels of oxidized cholesterol would break the contact and drive *Chrebp*α expression. This type of looping is known for other TFs for example, GATA-1 and its co-factor FOG-1. Both factors are required and need to physically interact to produce a DNA loop bringing the β-globin locus control region (LCR) together with the promoter [90]. Unbiased chromatin capturing techniques [86,91] is warranted to identify LXR-ChREBP trans-coactivation-dependent 3D chromatin contacts.

LXRα and ChREBPα promote glycolytic and lipogenic gene expression in high glucose conditions [3]. High cholesterol in the form of oxysterols, on the other hand, will activate LXR and drive cholesterol metabolism and efflux [92]. In the liver acetyl-CoA generated from β-oxidation of saturated fatty acids (SFAs) is used in cholesterol biosynthesis. Thus, diets rich in SFAs and trans-FAs might cause excess levels of cholesterol that need to be excreted via the bile [93]. While exogenous cholesterol contributes to the total, high cholesterol levels are often a result of or secondary to, high levels of dietary SFAs. This puts some of our observations in an interesting perspective. In the case of a carbohydrate and lipid-rich condition, the liver receives both glucose and oxysterol cues, activating both glucose-driven lipogenesis and oxysterol-driven cholesterol excretion. LXRα, which activates both pathways, also functions as a molecular switch (Figure 7). To prevent toxic accumulation of free cholesterol, de novo synthesized FAs is used to esterify cholesterol that are released as neutral cholesteryl esters in apoB-containing lipoproteins [94,95]. However, our data suggest that ligand-bound LXR will reduce the co-activation of glycolytic and lipogenic ChREBP target genes. Accordingly, LXRα might safeguard the liver against ectopic SFA levels. Alternatively, LXRα’s activation-to-repression relay is part of a negative feedback mechanism, ensuring that the activation of ChREBPα, co-activation of *Chrebpβ* transcription and subsequent induction of glycolytic and lipogenic genes is limited, when the local concentration of pyruvate, acetyl-CoA and SFAs, as well as cholesterol/oxysterol reaches a certain level. This hitherto unappreciated role of LXRα may have been overlooked due to LXR’s DNL-promoting function.

Several LXR-activating drugs have been developed through the years aiming at boosting reverse cholesterol transport [92,96]. However, the use of such agonists has been hampered by their lipogenic effects, leading to hypertriglyceridemia and hepatic steatosis [8,25]. As LXR seem to have both lipogenic-promoting and lipogenic-limiting effects, it is tempting to speculate if it is possible to separate the two. The success of such an effort would most probably lie in targeting both the DBD and LBD of LXRα, breaking the DNA:DBD contact [97,98] and activating the LBD. Preliminary in vitro data using the LXR DBD-mutant in combination with GW3965 suggest that this might be a possible approach.

## Figures and Tables

**Figure 2 cells-09-01214-f002:**
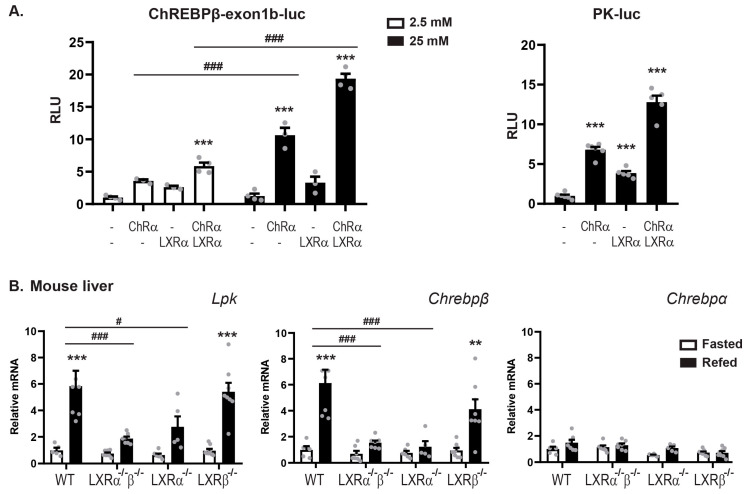
LXRα co-activates ChREBP specific target genes in vitro and in vivo. (**A**). Huh7 cells cultured in 25 mM glucose were transfected with a *Chrebpβ* (*n* = 3) or *Lpk*-driven luciferase reporter (*n* = 6) and plasmids expressing ChREBPα/Mlxγ, with or without LXRα/RXRα. The Renilla luciferase reporter pRL-CMV was used as internal control. Six hours post transfection, cells were treated with 2.5 mM or 25 mM glucose for 18 h. Dual luciferase reporter assays were performed 24 h post transfection. (**B**). LXRαβ wild type (WT), LXRα^-/-^, LXRβ^-/-^ and LXRα^-/-^β^-/-^ mice were fasted for 24 h (white bars) or fasted for 24 h and refed for 12 h (black bars) (*n* = 5–8 mice per group). Hepatic gene expression of *Lpk* (*Pklr*), *Chrebpβ* and *Chrebpα* was analyzed by quantitative RT-PCR, normalized to *Tbp* and the control group set to 1. For the WT fasted group, Ct values were ≈ 28 for *Chrebpα* and 29 for *Chrebpβ*. Data are presented as mean ± standard error of the mean (SEM). Significant differences are shown as ** *p* < 0.01, *** *p* < 0.001 compared to control within the same treatment and ^#^
*p* < 0.05, ^###^
*p* < 0.001 between indicated groups.

**Figure 3 cells-09-01214-f003:**
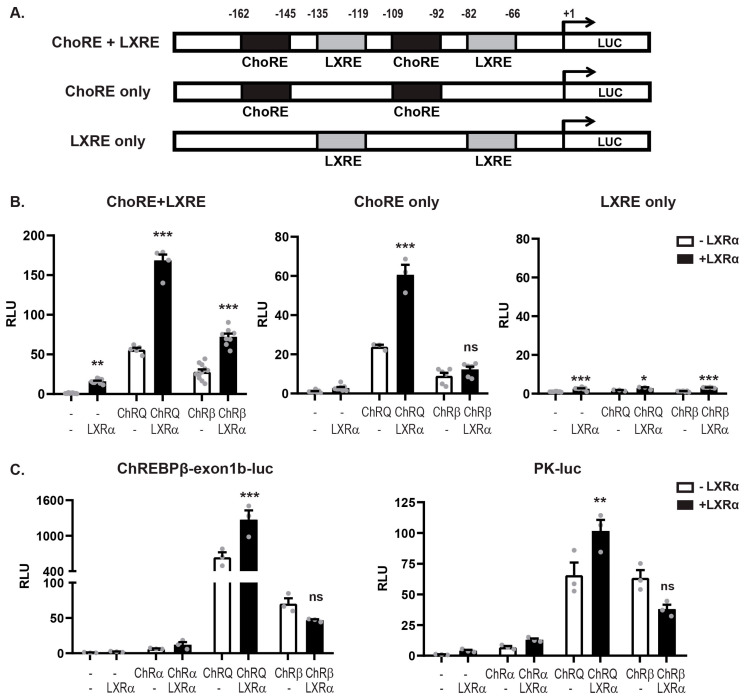
LXRα:ChREBPα co-activation requires functional ChoREs but not LXREs. (**A**). Schematic representation of the synthetic luciferase reporters constructs. ChoRE, carbohydrate response element; LXRE, LXR response element. (**B**). Huh7 cells cultured in 25 mM glucose were transfected with synthetic luciferase reporters containing ChoRE+LXRE, ChoRE-only or LXRE-only and plasmids expressing ChREBPα, the activated quadruple mutant ChREBP-Q or ChREBPβ, together with Mlxγ, with or without LXRα/RXRα. The Renilla luciferase reporter pRL-CMV was used as internal control. Dual luciferase reporter assays were performed 24 h post transfection. (**C**). Huh7 cells cultured in 25 mM glucose were transfected with a *Chrebpβ* or *Lpk*-driven luciferase reporter luciferase reporter and plasmids expressing ChREBPα, ChREBP-Q, ChREBPβ together with Mlxγ, with or without LXRα/RXRα. The Renilla luciferase reporter pRL-CMV was used as internal control. Dual luciferase reporter assays were performed 24 h post transfection. Data are presented as mean ± SEM (*n* = 3–7). Significant differences are shown as * *p* < 0.05, ** *p* < 0.01, *** *p* < 0.001 compared to control within the same ChREBP isoform transfection. ns, not significant.

**Figure 4 cells-09-01214-f004:**
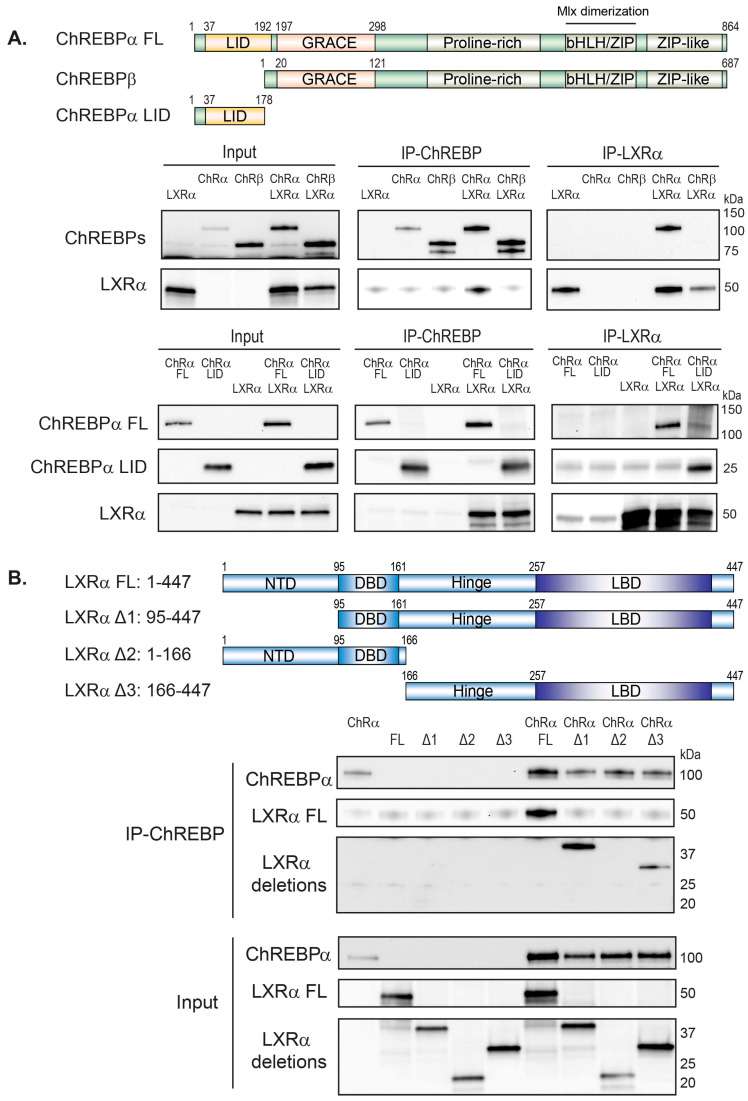
LXRα and ChREBPα interact via key activation domains. (**A**). Top panel: Schematic representation of the ChREBPα full-length (FL), ChREBPβ and the low glucose inhibitory domain (LID) protein. Bottom panel: CoIP of LXRα and ChREBPα, ChREBPβ or LID, expressed in COS-1 cells cultured in 25 mM glucose. The ChREBP expression plasmids were transfected with a DNA ratio of ChREBPα:ChREBPβ:LID = 1:6:1, to obtain comparable protein levels. Lysates were immunoprecipitated with ChREBP, FLAG (for LID) or LXRα antibodies and input and immunoprecipitated proteins immunoblotted with the same antibodies (*n* = 3). One representative western blot is shown. LID, low-glucose inhibitory domain; GRACE, glucose-response activation conserved element; bHLH, basic helix-loop-helix domain: ZIP, leucine zipper. (**B**). Top panel: Schematic representation of the LXRα FL and truncations. Bottom panel: CoIP of ChREBPα and LXRα FL or truncations expressed in COS-1 cells cultured in 25 mM glucose. Lysates were immunoprecipitated with ChREBP antibody (*n* = 3). Input and immunoprecipitated proteins were immunoblotted with ChREBP or FLAG (for LXRα FL and truncations) antibodies. One representative western blot is shown. NTD, N-terminal domain; DBD, DNA-binding domain; LBD, ligand-binding domain.

**Figure 5 cells-09-01214-f005:**
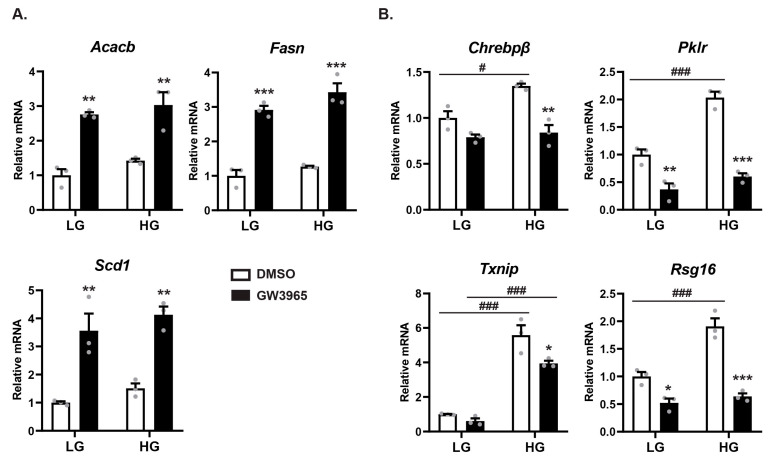
Ligand-activated LXR represses ChREBPα activity on ChREBP-specific target genes. Mouse primary hepatocytes were isolated and cultured in either 1 mM glucose (LG) or 25 mM glucose (HG) for 24 h. For the last 18 h the cells were treated with either DMSO (0.1%) or GW3965 (10 µM). Expression of (**A**) DNL genes *Acacb*, *Fasn*, *Scd1* and (**B**) ChREBP-specific target genes *Chrebpβ* (*Mlxiplβ)*, *Lpk* (*Pklr*), *Txnip* and *Rgs16* was analyzed by quantitative RT-PCR, normalized to *Tbp* and the control group set to 1. The Ct value for *Chrebpβ* was ≈ 26 for the LG DMSO treatment. Data are presented as mean ± SEM (*n* = 3). Significant differences are shown as * *p* < 0.05, ** *p* < 0.01, *** *p* < 0.001 compared to DMSO within the same glucose treatment and ^#^
*p* < 0.01, ^###^
*p* < 0.001 between LG and HG groups.

**Figure 6 cells-09-01214-f006:**
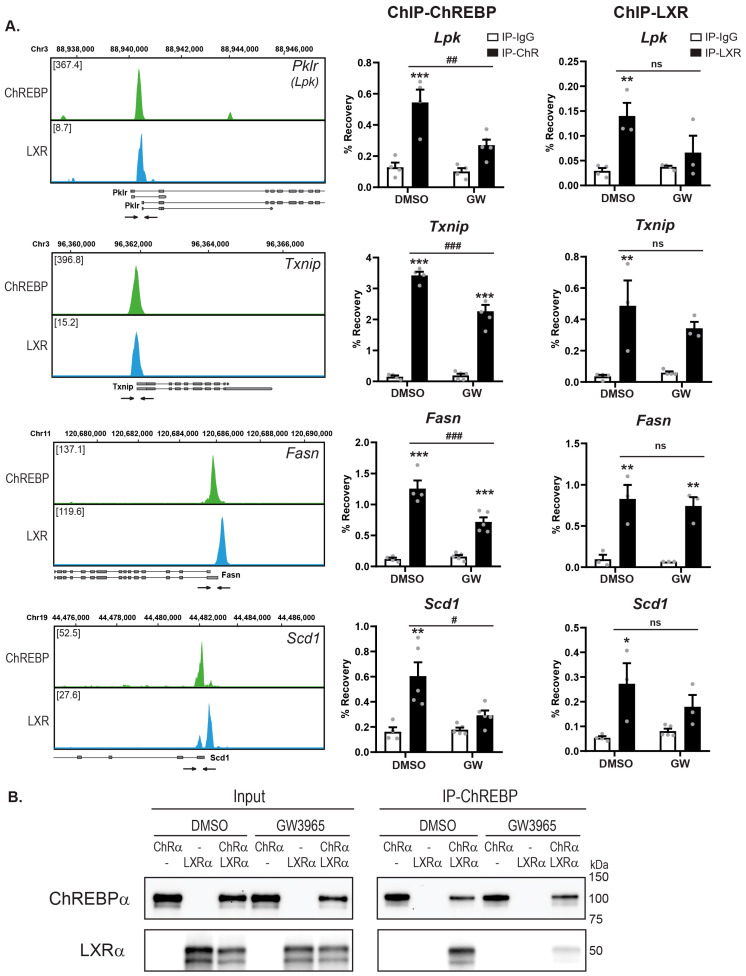
**Ligand-activated LXR reduces ChREBP binding to chromatin.** (**A**). Left panels: Local pattern of LXR-ChREBP co-occupancy. Browser view of LXR and ChREBP tracks in the promoter region of *Lpk* (*Pklr*), *Txnip*, *Fasn* and *Scd1*. Square brackets indicate the scale maxima of ChIP/input ratios. Arrows indicate the genomic locations of quantitative RT-PCR primers. Right panel: AML12 cells transfected with ChREBPα/Mlxγ and LXRα/RXRα were treated with DMSO (0.1%) or GW3965 (10 µM) for 18 h. ChREBP or LXR binding to genomic location indicated in the right panels were detected by ChIP using antibodies against ChREBP, LXR or IgG as negative control. Data are presented as mean ± SEM (*n* = 3–5). Significant differences are shown as * *p* < 0.05, ** *p* < 0.01, *** *p* < 0.001 compared to ChIP-IgG and ^#^
*p* < 0.05, ^##^
*p* < 0.01, ^###^
*p* < 0.001 between DMSO and GW3965 groups. ns, not significant. (**B**). CoIP of LXRα and ChREBPα, expressed in COS-1 cells cultured in 25 mM glucose, treated with DMSO (0.1%) or GW3965 (1 µM) for 18 h. Lysates were immunoprecipitated with ChREBP antibody (*n* = 3). Input and immunoprecipitated proteins were immunoblotted with ChREBP or LXRα antibodies. One representative western blot is shown.

**Figure 7 cells-09-01214-f007:**
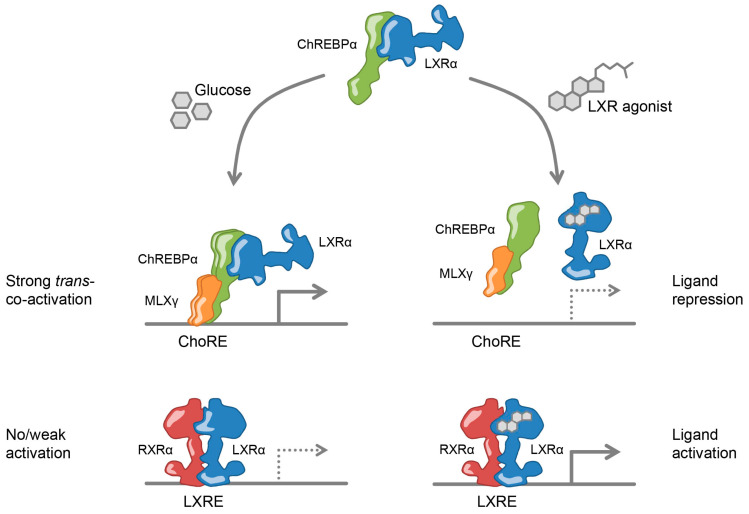
Model of transcriptional co-regulation by LXRα and ChREBPα.

**Table 1 cells-09-01214-t001:** Top 10 pathways enriched in genes with ChREBP-LXR peak pairs.

Pathway Name	Reactome ID	Candidates	*p*-Value	LXR	ChREBP
Metabolism	1430728	302 (15.4%)	3.6 × 10^−27^	**×**	**×**
Metabolism of lipids	556833	113 (17.1%)	4.6 × 10^−13^	**×**	**×**
Metabolism of amino acids and Derivatives	71291	61 (18.0%)	2.2 × 10^−8^		
Glucose metabolism	70326	24 (26.7%)	3.9 × 10^−7^	**×**	**×**
γ-carboxylation, transport and N- terminal cleavage of proteins	159854	8 (72.7%)	3.9 × 10^−7^		
Formation of Fibrin Clot	140877	14 (35.9%)	2.2 × 10^−6^		
Fatty acid metabolism	8978868	36 (19.5%)	2.7 × 10^−6^	**×**	**×**
γ-carboxylation of protein precursors	159740	7 (70.0%)	3.3 × 10^−6^		
Removal of N-terminal propeptides from γ-carboxylated proteins	159782	7 (70.0%)	3.3 × 10^−6^		
Gluconeogenesis	70263	12 (35.3%)	1.5 × 10^−5^	**×**	**×**

Gene filter: promoter to first exon (−5000 to +2500 bp), peak-to-peak distance <100 bp. Symbol **×** indicates pathways reported to be regulated by LXR and ChREBP in the literature.

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
