# Peer review of "LXRα Regulates ChREBPα Transactivity in a Target Gene-Specific Manner through an Agonist-Modulated LBD-LID Interaction"

_cells, 2020, doi:10.3390/cells9051214_

Round 1

Reviewer 1 Report

In the present study the author provide a comprehensive analysis of LXR and ChREBP transactivity and provide a new mechanistic model for transcriptional co-regulation. The presented datasets are thorough and rigorous and the overall results sound. I have only minor comments. First, in the introduction, the authors refer to LXR as a “cholesterol sensor” given its ability to be modulated by oxidized cholesterol derivatives, while this is true I find more appropriate to label LXR as a “nutritional” or “metabolic sensor”. As the authors also acknowledge, LXR activity also responds, directly or indirectly to other cues, such as glucose, for instance. Also, and although controversial, a previous study suggested glucose as a direct ligand of LXRs, the authors should briefly address this issue.  

The authors should provide the obtained PCR efficiency values and R2; this can also be added to Table S2. They should also clarify, in the methods section, how the relative gene expression was calculated; did they use the Livak method (or 2-ΔΔCT method) which allows relative gene expression normalization using a reference gene?

Regarding transactivation results, RXR was generally co-transfected along with LXR: does RXR play a role in LXR/ChoRE mediated responses?

The last sentence of the discussion sounds a bit too colloquial, in my opinion.

Line 480: replace semi-colon by colon

Line 525: erase “that”

Reviewer 3 Report

Title: LXRα regulates ChREBPα transactivity in a target gene-specific manner through an agonist-modulated LBD-LID interaction

Summary: The manuscript under review aims to investigate a possible mechanistic interplay between LXR-alpha and ChREBP-alpha and its  role in pathological conditions such as hepatic steatosis and insulin resistance. The idea to investigate such a link is novel and intriguing and represents a I am certain that the findings of this investigations will represent a high level of interest to the readers. The manuscript is well written, easy to follow and understand with clearly labeled figures with robust data provided and concise, clear content. The study is well designed, methods and statistical analyses are appropriate and up to date. Reference list is comprehensive and complete. Results are clearly explained and support the conclusions of the authors. I do not find any major concerns or deficiencies with the current manuscript; however, a typos check is warrant for identifying minor errors such as double periods or missing ones etc.

Round 2

Reviewer 2 Report

See attached file
